bioinformatics/genomics

adaption, bioinformatics, high altitude, long non-coding RNA, yak

**Author for correspondence:**
Qiu-Mei Ji
e-mail: jiqiumei07@163.com

[†]These authors contributed equally to this work.

# Transcriptome analysis identified long non-coding RNAs involved in the adaption of yak to high-altitude environments

Jin-Wei Xin[1,2,†], Zhi-Xin Chai[3,†], Cheng-Fu Zhang[1,2,†], Yu-Mei Yang[3], Qiang Zhang[1,2], Yong Zhu[1,2], Han-Wen Cao[1,2], Cidan Yang Ji[1,2], Jin-Cheng Zhong[3] and Qiu-Mei Ji[1,2]

[1]State Key Laboratory of Hulless Barley and Yak Germplasm Resources and Genetic Improvement, Lhasa, People's Republic of China
[2]Institute of Animal Science and Veterinary, Tibet Academy of Agricultural and Animal Husbandry Sciences, Lhasa, People's Republic of China
[3]Key Laboratory of Qinghai-Tibetan Plateau Animal Genetic Resource Reservation and Utilization, Sichuan Province and Ministry of Education, Southwest Minzu University, Chengdu, People's Republic of China

J-CZ, 0000-0002-1734-5887; Q-MJ, 0000-0003-3956-2484

The mechanisms underlying yak adaptation to high-altitude environments have been investigated using various methods, but no report has focused on long non-coding RNA (lncRNA). In the present study, lncRNAs were screened from the gluteus transcriptomes of yak and their transcriptional levels were compared with those in Sanjiang cattle, Holstein cattle and Tibetan cattle. The potential target genes of the differentially expressed lncRNAs between species/strains were predicted using *cis* and *trans* models. Based on *cis*-regulated target genes, no KEGG pathway was significantly enriched. Based on *trans*-regulated target genes, 11 KEGG pathways in relation to energy metabolism and three KEGG pathways associated with muscle contraction were significantly enriched. Compared with cattle strains, transcriptional levels of acyl-CoA dehydrogenase, acyl-CoA-binding protein, 3-hydroxyacyl-CoA dehydrogenase were relatively higher and those of glyceraldehyde 3-phosphate dehydrogenase, phosphoglycerate mutase 1, pyruvate kinase and lactate/malate dehydrogenase were relatively lower in yak, suggesting that yaks activated fatty acid oxidation but inhibited glucose oxidation and glycolysis. Besides, NADH dehydrogenase and ATP synthase

showed lower transcriptional levels in yak than in cattle, which might protect muscle tissues from deterioration caused by reactive oxygen species (ROS). Compared with cattle strains, the higher transcriptional level of glyoxalase in yak might contribute to dicarbonyl stress resistance. Voltage-dependent calcium channel/calcium release channel showed a lower level in yak than in cattle strains, which could reduce the $Ca^{2+}$ influx and subsequently decrease the risk of hypertension. However, levels of EF-hand and myosin were higher in yak than in cattle strains, which might enhance the negative effects of reduced $Ca^{2+}$ on muscle contraction. Overall, the present study identified lncRNAs and proposed their potential regulatory functions in yak.

## 1. Introduction

The Qinghai–Tibet Plateau, the highest plateau worldwide, has an extremely harsh environment. It is cold with low oxygen content and strong ultraviolet radiation [1]. Yak is the only large mammal in the Qinghai–Tibet Plateau and has genetically evolved phenotypical and physiological adaptation mechanisms to high-altitude environments, such as enhanced lung capacity, promoted oxygen delivery [2] and augmented endogenous nitric oxide production [3]. The pulmonary artery endothelial cell in yak is longer, wider and rounder than in cattle, which facilitates yak adaptation to high-altitude conditions [4].

The molecular mechanisms underlying yak adaptation to high-altitude environments have been explored extensively in recent years. Two whole genomes of Tibetan mammals have been sequenced to explore the molecular mechanisms underlying high-altitude adaptation [5,6]. Afterwards, several investigations at the mRNA level were conducted [7–9]. Moreover, two investigations focusing on the roles of microRNA (miRNA) in high-altitude adaptation have been reported. Guan *et al.* [10] revealed that differentially expressed (DE) miRNAs in heart and lung between yak and cattle enriched hypoxia-related pathways, including the HIF-1 signalling, insulin signalling, PI3 K-Akt signalling, nucleotide excision repair, cell cycle, apoptosis and fatty acid metabolism. Kong *et al.* [11] investigated changes in Jersey cattle in response to high-altitude hypoxia (HAH) compared with HAH-free condition. The results indicated that under HAH condition, Jersey cattle regulated inflammatory homeostasis by inhibiting the acute phase response, coagulation system, complement system and promoting liver X receptor/retinoid X receptor (LXR/RXR) activation. Three genes (*SLC1A2*, *HTT* and *SLC1A1*) encoding the glutamate receptor [12–14] were downregulated in the yak liver, suggesting that yak has reduced the import and transport of glutamate to reduce excitotoxicity, which could be induced by low oxygen condition and threat organisms [15,16]. Our transcriptome analysis [17] indicated that the transcriptional level of *BMPR2* was upregulated in yak heart and lung compared with in cattle, which might inhibit the proliferation of vascular smooth muscle [18,19] and thus suppress hypoxic pulmonary vasoconstriction. Moreover, *CHRNA3* and *SNCA* were upregulated in yak compared with cattle [17], which might promote the cardiac contractility of yak via neural and humoral regulation [20,21].

Muscle tissues require a large amount of oxygen. Responses of skeletal muscle metabolism to reduced oxygen availability are thought to influence physical capacity and systemic energy homeostasis in adaptation of animals to high-altitude environments [22]. Under high-altitude condition, exercise capacity drastically decreased in non-native animals, suggesting the weakened function of muscle tissues due to low oxygen [23]. Compared with cattle, yak's muscle shows higher activities of lactate dehydrogenase (LDH), malate dehydrogenase (MDH) and β-hydroxyacyl-CoA dehydrogenase (HOAD), displaying a higher anaerobic potential in carbohydrate metabolism and a higher oxidative capacity [24]. These results indicated that yak might develop special metabolism mechanisms in muscle tissues to adapt to high-altitude conditions. Moreover, our previous study compared transcriptome profiles of gluteus tissues between yak and low-altitude cattle strains. The results showed that yak differentially regulated mRNA expression of genes associated with immunity and blood coagulation in gluteus, which might facilitate their adaptation to high-altitude conditions [17].

Long non-coding RNAs (lncRNAs) play important regulatory roles at transcriptional, post-transcriptional, translational and epigenetic levels in variable cleavage, transcriptional interference, regulation of DNA methylation and protein modification [25–29]. Recent studies have also reported that lncRNAs participate in various physiological processes in bovines. Analysis of lncRNA expression in bovine macrophages suggested that lncRNAs regulated pathways of immune response during *Paratuberculosis* infection [30]. Four independent studies have proved the regulatory effects of lncRNA on proliferation and differentiation of skeletal muscle satellite cell in bovines [31–34]. Besides, Ma *et al.* [35] found that lncRNA *XIST* mediated inflammatory response via the NF-κB/NLRP3 inflammasome

pathway in bovine mammary epithelial cell. However, to the best of our knowledge, no report has investigated the roles of lncRNA in yak adaption to high-altitude conditions. To explore potential regulatory roles of lncRNA in yak, the present study examined DE lncRNAs in gluteus between yak and Sanjiang cattle, Tibetan cattle or Holstein cattle. Their regulatory effects on mRNA expression were predicted and the potential corresponding biological functions were discussed. These results provide new insights in relation to mechanisms underlying yak adaptation to high-altitude environments.

# 2. Material and methods

## 2.1. Sample preparation

The local farmers regularly kill dozens of Sanjiang cattle, Tibetan cattle, Holstein cattle and yak to sell meat. First, animals were anaesthetized using electrocution by attaching a pair of electrodes to the ears. Next, animals were killed by a lethal shock at 1 kV passing from ear to leg [36]. Blood was completely released before further processing. Most animals raised in the farms have clear record of birthday. From the batch of animals, female, healthy and 60-month old individuals with similar nutritional status were selected and fresh gluteus tissues were immediately collected *in vivo* after dissecting the skin at the slaughter house. Samples were frozen in liquid nitrogen until RNA isolation. For each species/strain, three replicates were prepared by collecting samples from three individuals. Dates and locations for sample collection have been described in Xin *et al.* [17].

## 2.2. RNA extraction, library preparation, sequencing and quality analysis

Total RNA was extracted from gluteus tissues using Biozol reagent (Bioer, Hangzhou, China), according to the manufacturer's protocol. Quality of RNA was examined using an Agilent Bioanalyzer 2100 system (Agilent Technologies, CA, USA). An RNA integrity number (RIN) higher than 8.0 was considered qualified. The quantity of RNA was measured using the Qubit RNA assay kit on a Qubit 3.0 Flurometer (Life Technologies, CA, USA).

For each sample, 3 µg of total RNA was used to prepare the sequencing library. Firstly, ribosomal RNA was removed using an Epicentre Ribo-zero rRNA Removal kit (Epicentre, USA), which was then cleaned up by ethanol precipitation. Next, sequencing libraries were generated using a NEBNext Ultra Directional RNA Library Prep Kit for Illumina (NEB, USA). Index-coded samples were clustered on a cBot cluster generation system using a HiSeq 4000 PE cluster kit (Illumina). Afterwards, the libraries were sequenced using an Illumina Hiseq 4000 platform to collect 150 bp paired-end reads.

FastQC v.0.11.8 (http://www.bioinformatics.babraham.ac.uk/projects/fastqc/) was used to evaluate the quality of raw data [37]. Reads having more than 1% unknown bases, reads containing adapters and reads with low-quality bases (with greater than 50% bases having a Phred quality score less than or equal to 15) were removed. Simultaneously, indices of clean reads, including Q20, Q30 and GC contents were calculated. All subsequent analyses were based on the clean data.

## 2.3. Annotation of unigenes and identification of lncRNAs

Clean reads were mapped to the reference genome of yak (BioProject number in GenBank: AGSK00000000) using the STAR alignment program v. 2.5.1b [38]. The mapped reads with mismatches less than 5 bp were assembled and transcripts were quantified using StringTie package v. 1.3.4 [39].

LncRNAs were identified according to the following workflow. Firstly, the class-code of transcripts were identified using Cuffcompare package, a tool of Cufflink suite (http://cole-trapnell-lab.github.io/cufflinks/cuffcompare/). Transcripts belonging to classes i, j, x, u and o were retained. Next, transcripts with length less than 200 nt, reads count less than 20 and/or fragments per kilobase of transcript sequence per million base pairs sequenced (FPKM) value less than 20 were removed. Finally, non-coding transcripts were selected using the coding-non-coding-index (CNCI) [40] and coding potential calculator (CPC) tools [41]. The non-coding transcripts identified by both CNCI and CPC tools were candidate lncRNAs.

## 2.4. Comparison of mRNA and lncRNA expression levels

The union model of HTseq v. 0.60 [42] was used to calculate FPKM values of each unigene and lncRNA in each sample. Pairwise comparison of FPKM values between two species/strains were conducted using

DESeq2R package v. 3.8 [43]. Differences with false discovery rate (FDR) less than 0.05 (using the BH method; [44]) and fold change greater than 2 were considered statistically significant.

## 2.5. Target gene prediction of lncRNAs

The target genes of lncRNAs were predicted in both *cis* and *trans* models. For the prediction of *cis* target genes [45], the coding genes within 100 kb upstream and downstream from the location of lncRNA were retrieved. *Trans* regulation is not dependent on positional relationship. *Trans* regulation of lncRNAs in the present study was predicted by calculating the binding energies using RNAplex [46]. The parameters for RNAplex were set as $-e < -20$ and target genes localized to the same chromosome of the lncRNA were removed [47].

The identified target genes were mapped to gene ontology (GO) database and Kyoto Encyclopedia of Genes and Genomes database (KEGG) [48] for enrichment of GO categories and KEGG pathways using BLAST software [49]. The significance of GO term and KEGG pathway enrichment was examined using the Fisher's exact test. The *p*-values were corrected using the BH method [44] by setting the FDR < 0.05 to produce *Q* values. GO terms or KEGG pathways displaying *Q*-value < 0.05 were considered significantly enriched.

## 2.6. Validation of lncRNA expression levels using real-time quantitative PCR

Ten lncRNAs were randomly selected (electronic supplementary material, table S1) and their expression levels were validated using real-time qPCR (RT-qPCR). The cDNA was reverse transcribed using the BioRT cDNA first strand synthesis kit (Bioer, Hangzhou, China) with random primers according to the manufacturer's protocol. RT-qPCR experiments were carried out using BioEasy master mix (Bioer, Hangzhou, China) on a Line Gene9600 Plus qPCR machine (Bioer, Hangzhou, China). The sequences of primers are shown in electronic supplementary material, table S1. *β-actin* was used as the internal control. The relative expression levels of each lncRNA were calculated using the typical $2^{-\Delta\Delta Ct}$ method [50]. The relative transcription levels of each gene were compared statistically between species/strains using Student's *t*-test in SPSS 20.

# 3. Results and discussion

## 3.1. Identification of lncRNAs

The Illumina sequencing data have been deposited in GenBank with the BioProject number of PRJNA512958. After filtering, the number of total clean reads ranged from 75.4 to 170.8 million, and Q20 values ranged from 96.27% to 97.59% for all samples (electronic supplementary material, table S2). STAR alignment showed that 86.04–96.15% reads could map to the reference genome for each sample (electronic supplementary material, table S3).

In total, 1364 lncRNAs were identified and all of them were novel lncRNAs (unidentified in all species) with median length of 883 bp (electronic supplementary material, table S4). Approximately, 80.65%, 8.58% and 10.78% of lncRNAs were less than 2000 bp, 2000–3000 bp and greater than 3000 bp in length, respectively (figure 1). Sequences of the identified lncRNAs are shown in electronic supplementary material.

## 3.2. DE lncRNAs and qPCR validation

In order to validate expression levels of lncRNAs, 10 DE lncRNAs were selected for qPCR. Overall, the qPCR and FPKM results showed similar tendencies, suggesting the reliability of levels calculated by FPKM values (figure 2). Pairwise comparisons between the four species/strains revealed that the number of DE lncRNAs in comparison between yak and Tibetan cattle (193) was smaller than that between yak and Sanjiang cattle (361), and between yak and Holstein cattle (433) (table 1). The FPKM values of all DE lncRNAs and statistical analyses results are included in the electronic supplementary material.

The clustering analysis of DE lncRNAs between the four samples displayed two clusters. One included yak and Tibetan cattle, and the other contained Sanjiang and Holstein cattle (figure 3). This pattern was consistent with the clustering result of mRNA expression profiles [17]. Tibetan, Sanjiang and Holstein cattle belong to the same species. However, yak and Tibetan cattle live at relatively high

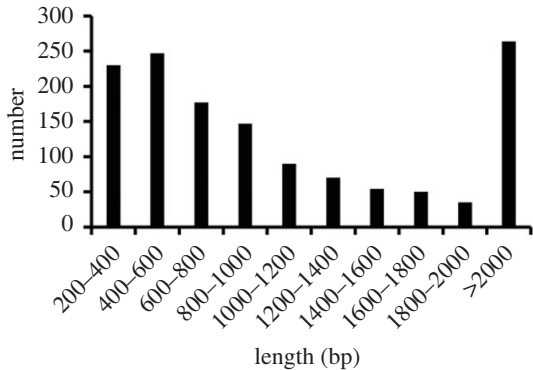

**Figure 1.** Length distribution of identified lncRNAs in gluteus.

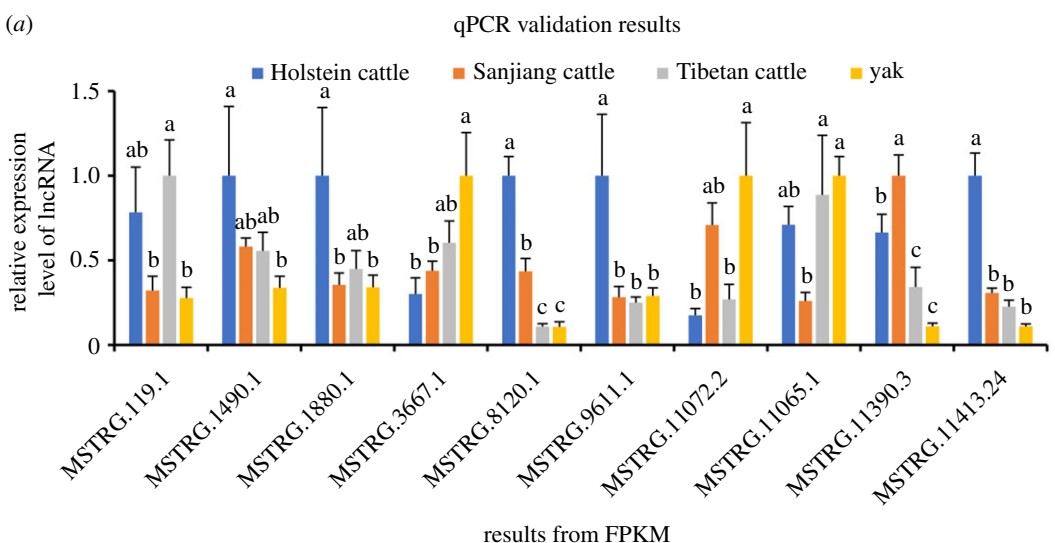

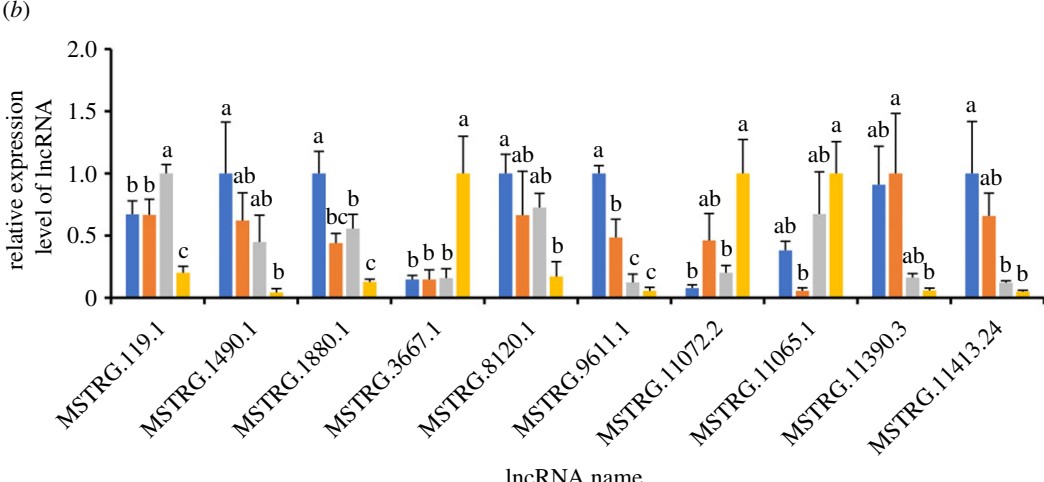

**Figure 2.** Validation of DE lncRNAs through qPCR. Different letters above bars indicate significant difference ($p < 0.05$).

altitudes, while Sanjiang and Holstain cattle live at low altitudes. The adaptation of Tibetan cattle to local environments might regulate lncRNA transcription in gluteus tissues, driving the separation of Tibetan cattle from Holstein and Sanjiang cattle on the clustering pattern. Besides, adaptive introgression has been reported in butterflies [51] and humans [52], and significant gene flow from yak to Tibetan cattle has been detected [53–55]. Adaptive introgression from yak probably also took place in Tibetan cattle, which may drive the clustering of yak and Tibetan cattle.

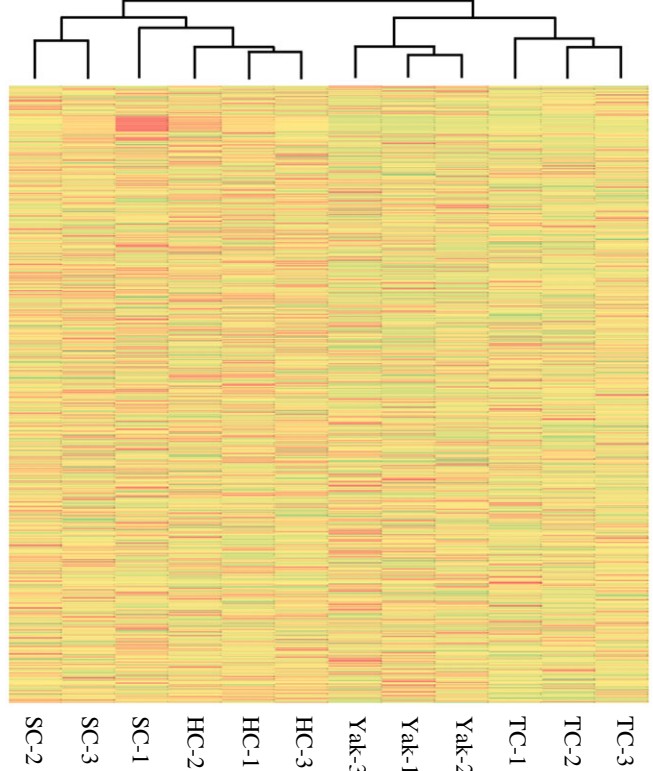

**Figure 3.** Clustering patterns of DE lncRNAs among yak, Sanjiang cattle (SC), Holstein cattle (HC) and Tibetan cattle (TC).

**Table 1.** Numbers of DE lncRNA in gluteus. SC, Sanjiang cattle; HC, Holstein cattle; TC, Tibetan cattle.

|      | SC  | HC  | TC  | yak |
|------|-----|-----|-----|-----|
| SC   | —   | 82  | 361 | 101 |
| HC   | 82  | —   | 433 | 211 |
| TC   | 361 | 433 | —   | 193 |
| Yak  | 101 | 211 | 193 | —   |

## 3.3. Target prediction of lncRNAs and KEGG enrichment

Based on *cis*-regulated target genes, KEGG enrichment analysis revealed no significantly enriched pathway between yak and any cattle strain. Differentially, based on *trans*-regulated target genes, KEGG enrichment analysis significantly revealed 4, 15, 3 and 11 KEGG pathways in comparisons between yak and Sanjiang cattle, yak and Holsten cattle, Tibetan cattle and Sanjiang cattle, Tibetan cattle and Holsten cattle, respectively (all *Q*-values < 0.05; table 2). Among these pathways, 11 were related to energy metabolism (ko00010, ko00620, ko00190, ko04932, ko00020, ko01210, ko00071, ko01212, ko01230, ko04146 and ko04922), and three associated with muscle contraction (ko05410, ko05414 and ko04260). In high-altitude habitats which are characterized by low temperature and low oxygen content, yak demands high metabolism to maintain body temperature. On the one hand, yak would produce more heat by muscle contraction after acclimation to cold environments [56,57]. On the other hand, for adaptation to low oxygen condition, yak needs to increase the efficiency of $O_2$ utilization. As previously reported, mice showed a higher efficiency of $O_2$ utilization under low oxygen conditions [58]. Thus, enrichment of energy metabolism-related and muscle contraction-related pathways in the present study could be attributed to yak adaptation to high-altitude environments.

For further analysis, the expression levels of 130 target genes in relation to energy metabolism and 37 target genes associated with muscle contraction were retrieved from our previously published transcriptome data [17]. Student's *t*-tests indicated that 25 genes involved in energy metabolism and

**Table 2.** Significantly enriched KEGG pathways of genes targeted by DE lncRNA between yak and Tibetan cattle, Holstein or Sanjiang cattle. The *p*-values indicate statistical significance and *Q*-values represent corrected *p*-values using Benjamini and Hochberg's method.

| KEGG ID | name of KEGG pathway | involved/total gene numbers | *p*-value | *Q*-value |
|---------|----------------------|------------------------------|-----------|-----------|
| Sanjiang cattle versus yak | | | | |
| ko00010 | glycolysis/gluconeogenesis | 11/271 | 0.00 | 0.01 |
| ko05410 | hypertrophic cardiomyopathy (HCM) | 12/271 | 0.00 | 0.01 |
| ko05414 | dilated cardiomyopathy | 11/271 | 0.00 | 0.04 |
| ko00620 | pyruvate metabolism | 7/271 | 0.00 | 0.04 |
| Holstein cattle versus yak | | | | |
| ko00190 | oxidative phosphorylation | 76/420 | 0.00 | 0.00 |
| ko04932 | non-alcoholic fatty liver disease (NAFLD) | 60/420 | 0.00 | 0.00 |
| ko04260 | muscle contraction | 30/420 | 0.00 | 0.00 |
| ko00020 | citrate cycle (TCA cycle) | 19/420 | 0.00 | 0.00 |
| ko00620 | pyruvate metabolism | 13/420 | 0.00 | 0.00 |
| ko01210 | 2-oxocarboxylic acid metabolism | 7/420 | 0.00 | 0.00 |
| ko00071 | fatty acid degradation | 10/420 | 0.00 | 0.00 |
| ko01212 | fatty acid metabolism | 11/420 | 0.00 | 0.00 |
| ko03050 | proteasome | 10/420 | 0.00 | 0.00 |
| ko01230 | biosynthesis of amino acids | 13/420 | 0.00 | 0.00 |
| ko05410 | hypertrophic cardiomyopathy (HCM) | 14/420 | 0.00 | 0.00 |
| ko00010 | glycolysis/gluconeogenesis | 12/420 | 0.00 | 0.01 |
| ko04146 | peroxisome | 13/420 | 0.00 | 0.01 |
| ko05414 | dilated cardiomyopathy | 13/420 | 0.00 | 0.03 |
| ko04922 | glucagon signalling pathway | 12/420 | 0.00 | 0.04 |
| Sanjiang cattle versus Tibetan cattle | | | | |
| ko05150 | *Staphylococcus aureus* infection | 8/120 | 0.00 | 0.00 |
| ko05140 | leishmaniasis | 7/120 | 0.00 | 0.00 |
| ko04380 | osteoclast differentiation | 8/120 | 0.00 | 0.01 |
| Holsten cattle versus Tibetan cattle | | | | |
| ko00190 | oxidative phosphorylation | 40/287 | 0.00 | 0.00 |
| ko04932 | non-alcoholic fatty liver disease (NAFLD) | 31/287 | 0.00 | 0.00 |
| ko04260 | muscle contraction | 22/287 | 0.00 | 0.00 |
| ko00020 | citrate cycle (TCA cycle) | 12/287 | 0.00 | 0.00 |
| ko00010 | glycolysis/gluconeogenesis | 13/287 | 0.00 | 0.00 |
| ko00620 | pyruvate metabolism | 10/287 | 0.00 | 0.00 |
| ko04020 | calcium signalling pathway | 21/287 | 0.00 | 0.00 |
| ko01230 | biosynthesis of amino acids | 11/287 | 0.00 | 0.00 |
| ko04922 | glucagon signalling pathway | 12/287 | 0.00 | 0.00 |
| ko05410 | hypertrophic cardiomyopathy (HCM) | 11/287 | 0.00 | 0.01 |
| ko05414 | dilated cardiomyopathy | 10/287 | 0.00 | 0.00 |
| Tibetan cattle versus yak | | | | |
| ko04145 | phagosome | 14/120 | 0.00 | 0.00 |
| ko05150 | *Staphylococcus aureus* infection | 8/120 | 0.00 | 0.00 |

(*Continued.*)

**Table 2.** (Continued.)

| KEGG ID | name of KEGG pathway | involved/total gene numbers | *p*-value | *Q*-value |
| --- | --- | --- | --- | --- |
| ko05140 | leishmaniasis | 7/120 | 0.00 | 0.00 |
| ko05133 | pertussis | 7/120 | 0.00 | 0.00 |
| ko04380 | osteoclast differentiation | 8/120 | 0.00 | 0.01 |

11 genes participating in muscle contraction were significantly differentially expressed in at least one comparison between four samples (electronic supplementary material, table S5). To further discuss the biological functions of these genes, the transcriptional levels of 11 coding genes related to energy metabolism pathways and 7 coding genes related to muscle contraction pathways were compared (figures 4 and 5), and their biological functions were discussed.

## 3.4. Regulation of energy metabolism in yak skeletal muscle

Genetic selection on the peroxisome proliferator-activated receptor gamma coactivator 1-alpha (*PPARA*) gene has been reported in Tibetan human populations, suggesting that altered fatty acid (FA) metabolism might be a feature of long-term adaptation to high altitude [59]. It is generally accepted that exposure to low oxygen condition induces a selective attenuation of FA oxidation, while glucose uptake is maintained or increased [60]. In the present study, the transcriptional levels of three DE lncRNAs (MSTRG.16170.2, MSTRG.23394.1 and MSTRG.26388.4) were highly correlated with the levels of BmuPB003110, BmuPB012824 and BmuPB017956 all encoding FA oxidative enzyme Acyl-CoA dehydrogenase ($R^2 = 0.913$, 0.906 and 0.914, respectively). These results suggested that these three lncRNAs might target on FA oxidative enzyme Acyl-CoA dehydrogenase. The DE lncRNA MSTRG.13013.1 might target on acyl-CoA-binding protein (BmuPB008201) with $R^2$ of 0.915, and the DE lncRNA MSTRG.26388.4 might target on 3-hydroxyacyl-CoA dehydrogenase (BmuPB021264) with $R^2$ of 0.913. These five lncRNAs and their targeted coding genes all showed significantly higher transcriptional levels in yak compared with Holstein cattle (figure 4*a* and electronic supplementary material, table S5), suggesting that FA oxidation process was more activated in yak than in Holstein cattle. Besides, the transcriptional levels of five enzymes involved in glycolysis including glyceraldehyde 3-phosphate dehydrogenase (BmuPB000082, possibly a target of MSTRG.24686.4, $R^2 = 0.926$), phosphoglycerate mutase 1 (BmuPB012923, possibly a target of MSTRG.24686.4, $R^2 = 0.908$), pyruvate kinase (BmuPB009427, possibly a target of MSTRG.24686.4, $R^2 = 0.920$) and lactate/MDH (BmuPB021081 and BmuPB017171, possibly a target of MSTRG.11882.2 and MSTRG.24686.4 with $R^2$ of 0.907 and 0.924, respectively) were lower in yak compared with Holstein cattle, Sanjiang cattle and Tibetan cattle (figure 4*b* and electronic supplementary material, table S5). These results suggested that yak might preferentially use lipids for metabolism. Similar results were observed in comparison between Tibetan people (native to high-altitude condition) and Nepali people (permanently residing at a low altitude) [61]. Lipids make up more than 80% of the total energy reserve in mammals, and their energy density is an order of magnitude greater than that of carbohydrates [62]. Thus, lipids are a preferred fuel source that offsets the rapid depletion of carbohydrates in response to the combined stresses of low oxygen and low temperature.

NADH dehydrogenase is the first and largest enzyme complex in the respiratory chain and acts as a proton pump. ATP synthase, the last enzyme in the respiratory chain, couples with the mitochondrial inner membrane electrochemical gradient to synthesize ATP. In the present study, MSTRG.23993.3 and its potential target NADH dehydrogenase (BmuPB009505, $R^2 = 0.926$) showed significantly lower levels in yak compared with Holstein cattle, Sanjiang cattle and Tibetan cattle. MSTRG.5970.1 and its potential target ATP synthase (BmuPB005211, $R^2 = 0.916$) showed significantly lower levels in yak than those in Holstein cattle (figure 4*c* and electronic supplementary material, table S5). The downregulation of respiratory enzymes may contribute to yak adaptation to high-altitude environments. The decreased respiration not only facilitated oxygen utilization under insufficient oxygen conditions but also protected muscle tissues from deterioration caused by reactive oxygen species (ROS) [63].

Dicarbonyl stress is the abnormal accumulation of a-oxoaldehyde metabolites (methylglyoxal, glyoxal and 3-deoxyglucosone), which is harmful to protein and DNA and may induce cell and tissue dysfunction, ageing and disease [64]. Both hypoxia and increased glucose metabolism could induce dicarbonyl stress [65]. Glyoxal and methylglyoxal are metabolized mainly by glyoxalase. In the

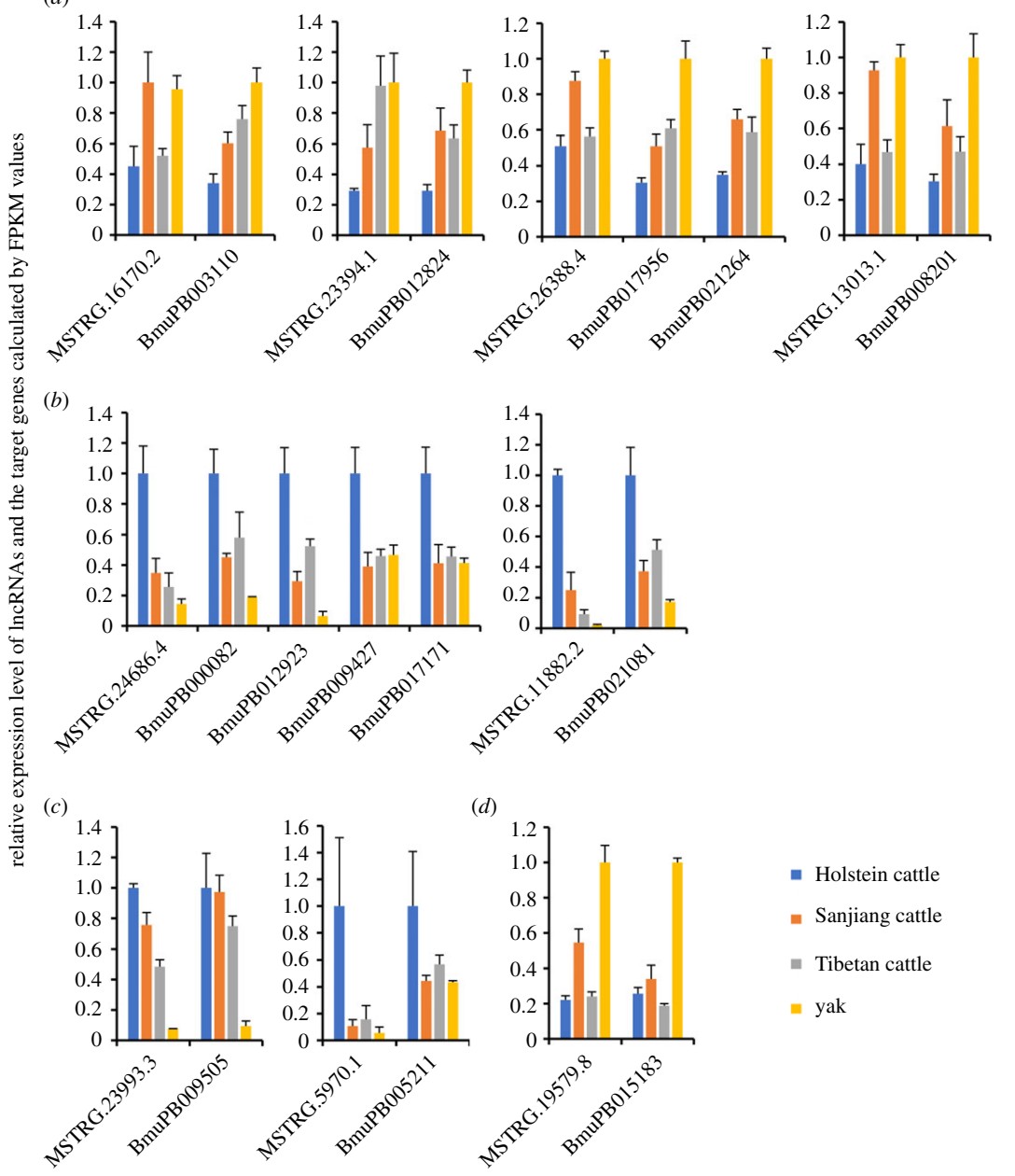

**Figure 4.** Relative transcriptional levels of differentially expressed lncRNAs and their target genes in relation to energy metabolism calculated by FPKM values.

present study, compared with Sanjiang, Holstein and Tibetan cattle, the transcriptional levels of MSTRG.19579.8 and its potential target glyoxalase (BmuPB015183, $R^2 = 0.959$) increased significantly in yak (figure 4d and electronic supplementary material, table S5), which may improve yak's resistance to dicarbonyl stress under low oxygen conditions.

## 3.5. Regulation of muscle contraction in yak

It has been well demonstrated that exposure to low oxygen condition increases glucose consumption in animal muscle tissues [66], which is regulated by increased functions of $Ca^{2+}$-pumps [67]. In high-altitude environments, animals are always exposed to low oxygen conditions. Constant high level of cytosolic $Ca^{2+}$ in muscle cells would induce hypertension in these animals [68]. Yak is a native species to high-altitude environments. Thus, they should have evolved regulatory mechanisms to avoid

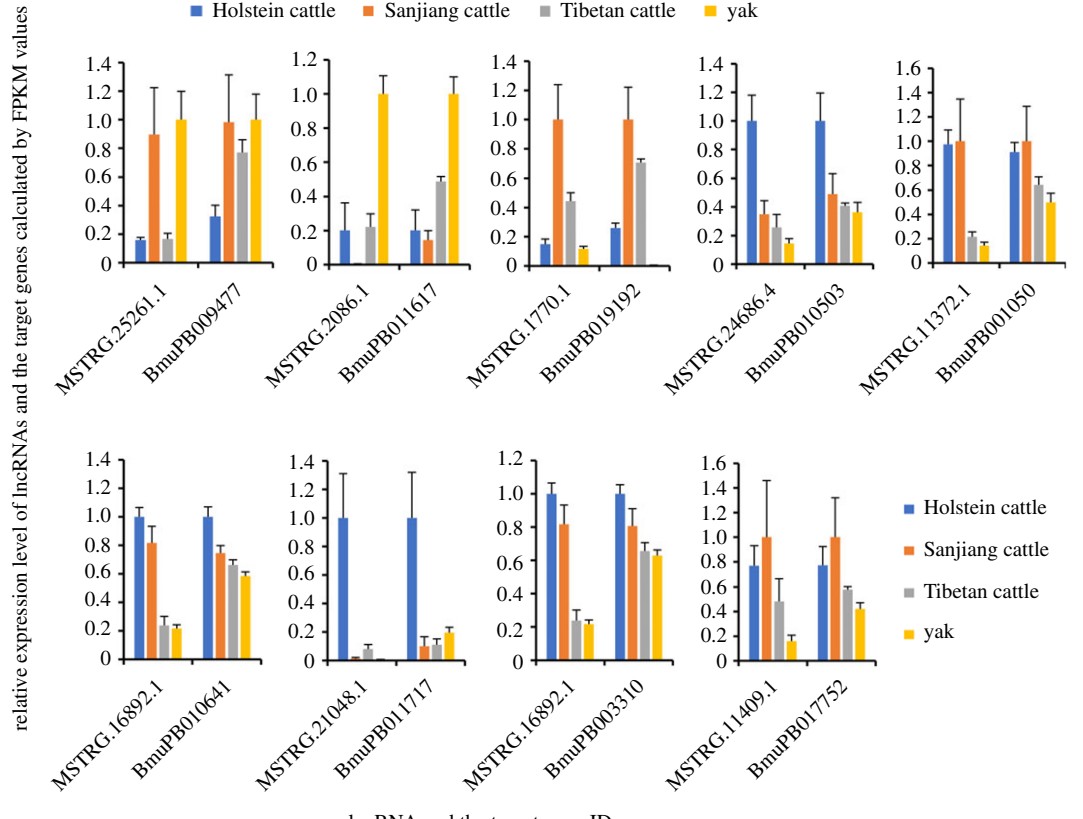

**Figure 5.** Relative transcriptional levels of differentially expressed lncRNAs and their target genes in relation to muscle contraction calculated by FPKM values.

hypoxia-induced high $Ca^{2+}$ level. In the present study, the transcriptional levels of lncRNAs MSTRG.16892 and MSTRG.21048.1 were highly correlated to two unigenes encoding voltage-dependent calcium channel (BmuPB010641, $R^2 = 0.904$; BmuPB011717, $R^2 = 0.958$; respectively). Similarly, the transcriptional level of MSTRG.11372.1 was positively correlated to calcium release channel (BmuPB001050, $R^2 = 0.909$). These results suggested that transcription of $Ca^{2+}$ pumps might be regulated by lncRNAs in yak muscles. More importantly, all these transcripts showed significantly lower levels in yak compared with the three cattle strains (figure 5 and electronic supplementary material, table S5) might inhibit $Ca^{2+}$ transportation and minimize the potential harms caused by hypoxia. Moreover, sarcoglycan is a component of the dystrophin–glycoprotein complex, which plays a role in the maintenance of muscle cell integrity by binding to multiple basement membrane proteins and forming a transmembrane link to the actin cytoskeleton [69]. It has been reported that sarcoglycan functions in calcium homeostasis in skeletal muscle fibres [70]. The present data also revealed significantly lower transcriptional levels of sarcoglycan (BmuPB003310, potentially targeted by MSTRG.16892.1, $R^2 = 0.911$, figure 5 and electronic supplementary material, table S5) in yak than in other animals, which might also contribute to the $Ca^{2+}$ homeostasis in yak muscles.

Besides the functions in glucose uptake, $Ca^{2+}$ is also greatly important to muscle contraction. When an action potential is generated, voltage-dependent calcium channel/calcium release channel on the sarcoplasmic reticulum (SR) is activated, which next releases $Ca^{2+}$ from SR into the sarcoplasm and initiates skeletal muscle contraction. When $Ca^{2+}$ pump transfers $Ca^{2+}$ into SR, muscles relax. Forces are generated in striated muscles from the cyclical interaction between myosin and actin, which is mediated by the actin-associated regulatory proteins, troponin and tropomyosin. In the absence of $Ca^{2+}$, tropomyosin sterically prevents myosin from binding to actin; while upon $Ca^{2+}$ binding to troponin through a pair of EF-hand [71], tropomyosin's equilibrium position shifts, allowing cooperative binding of myosin by exposing neighbouring actin binding sites [72,73]. To minimize the negative effects of decreased levels of $Ca^{2+}$ pumps, yak may regulate other components during muscle contraction. In the present study, the transcriptional levels of myosin (BmuPB009477) and EF-hand domain (BmuPB011617) were highly correlated with the lncRNAs MSTRG.25261.1 and MSTRG.2086.1

with $R^2$ of 0.916 and 0.931, respectively, suggesting the potential regulatory relationship between them. Their transcriptional levels all increased significantly in yak, compared with other animals (figure 5 and electronic supplementary material, table S5). Over-representation of these genes should increase the binding ability to $Ca^{2+}$ and subsequently enhance muscle contraction even at a low level of $Ca^{2+}$. The transcription of tropomyosin (BmuPB019192 and BmuPB010503) might be regulated by lncRNAs MSTRG.1770.1 and MSTRG.24686.4, respectively, since the $R^2$ between their transcriptional levels were high (0.946 and 0.929, respectively). Tropomyosin showed significantly lower level in yak compared with the three cattle strains, which might facilitate the muscle contraction, since tropomyosin plays negative roles during this process.

Moreover, titin is responsible for the elasticity of striated muscle by providing connections between microfilaments [74]. In the present study, titin (BmuPB017752) was positively related to the transcriptional level of MSTRG.11409.1 ($R^2 = 0.913$), demonstrating a potential regulatory relationship. Both transcripts showed significantly lower level in yak than those in cattle strains (figure 5 and electronic supplementary material, table S5). These changes may have biological significances to yak adaption to the high-altitude environments. Similar results have been reported. Expression level of titin was significantly reduced in response to a low oxygen condition in rats, which may associate with the decline of passive tension of diaphragm [75].

# 4. Conclusion

The present study identified 1365 lncRNAs from the transcriptome of yak muscle tissues. Compared with yak, 193, 361 and 433 lncRNAs were significantly differentially expressed in Tibetan cattle, Sanjiang cattle and Holstein cattle, respectively. The potential target genes of these DE lncRNAs were predicted, which might regulate energy metabolism and muscle contraction in yak. These changes would promote yak adaptation to high-altitude environments.

Ethics. Samples used in the present study were purchased from the local farmers. The animals were sacrificed by the farmers following the method described in the Material and methods section. No special ethical statement is required for the present study.

Data accessibility. All data have been included in the manuscript and electronic supplementary material.

Authors' contributions. J.-W.X., Z.-X.C. and C.-F.Z. carried out the molecular laboratory work, participated in data analysis, carried out sequence alignments, participated in the design of the study and drafted the manuscript; Y.-M.Y., and Q.Z. carried out the statistical analyses and critically revised the manuscript; Y.Z., H.-W.C. and C.Y.J. collected field data and critically revised the manuscript; J.-C.Z. and Q.-M.J. conceived of the study, designed the study, coordinated the study and helped draft the manuscript. All authors gave final approval for publication and agree to be held accountable for the work performed therein.

Competing interests. We declare we have no competing interests.

Funding. This work was supported by a program of Provincial Department of Finance of the Tibet Autonomous Region (grant no. XZNKY-2019-C-052), the Key Research and Development Projects in Tibet: Preservation of Characteristic Biological Germplasm Resources and Utilization of Gene Technology in Tibet (grant no. XZ202001ZY0016N), the Second Tibetan Plateau Scientific Expedition and Research Program (STEP) (grant no. 2019QZKK0501), the Ring-fenced Funding of Finance Department of Tibet Autonomous Region, the earmarked fund for Modern Agro-industry Technology Research System (grant no. CARS-37), Basic Research Programs of Sichuan Province (grant no. 2019YJ0256) and the Open Project Program of State Key Laboratory of Hulless Barley and Yak Germplasm Resources and Genetic Improvement (Tibet Academy of Agricultural and Animal Husbandry and Animal Husbandry Sciences) (grant no. XZNKY-2019-C-007K10).

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
