## [Reviewer comments · Royal Society Open Science]

Review History

RSOS-200625.R0 (Original submission)

Review form: Reviewer 1

Is the manuscript scientifically sound in its present form?

Yes

Are the interpretations and conclusions justified by the results?

Yes

Is the language acceptable?

Yes

Do you have any ethical concerns with this paper?

No

Have you any concerns about statistical analyses in this paper?

No

Recommendation?

Accept with minor revision (please list in comments)

Comments to the Author(s)

Xin et al. explored their transcriptome sequencing data of yak and three cattle strains, identified lncRNAs and discussed their potential biological functions. To me, the major weakness of the present study is the lack of further validation of lncRNA's function. Under the trans model, false correlations might be included in your identification. However, I understand that yak is not a model organism and experiments on yak cell lines might be impossible currently. As a bioinformatic analysis, I think this manuscript can be published, because it provides novel lncRNA information for further research on yak. However, I encourage the authors to mention this weakness in the discussion part.

Throughout the whole manuscript, the authors used the word hypoxia instead of low oxygen condition. I think it is wrong. Hypoxia means the symptoms of organism under low oxygen condition.

Line 58: delete focused

Line 60: focusing on

Line 212: Why did you found none enriched pathways using the cis model? Is it normal in other animals? any mistake in your analysis process?

Line 215: comparisons

Line 292: it is inappropriate to using the word "uptake"

Line 294: what do you mean "consistent"?

Line 296: revised as "yak is a native species to high-altitude environments"

Review form: Reviewer 2

Is the manuscript scientifically sound in its present form?

Yes

Are the interpretations and conclusions justified by the results?

Yes

Is the language acceptable?

No

Do you have any ethical concerns with this paper?

Yes

Have you any concerns about statistical analyses in this paper?

Yes

Recommendation?

Major revision is needed (please make suggestions in comments)

Comments to the Author(s)

This article described the expression profile of lncRNAs in the gluteus tissues of Sanjiang cattle, Tibetan cattle, Holstein cattle and yak. And authors screened the potential functions and pathways of differential expression lncRNAs between yak and other cattle species. As a catalog of new transcripts, this contributes a portion of the available data. However, some problems with methods and data will affect data reliability.

Major:

Line 111 - In "2.1 Sample preparation", since the four species of cattle are raised by local farmers, apart from the differences in the transcriptome of the gluteus tissues caused by species, is it possible that the differences caused by nutritional status?

Line 113 - In "2.1 Sample preparation", please add related methods of gluteal muscle tissue collection after "All the involved animals were 60-month old". For example, whether it is collected in vivo, and how to treat the wound after collection. If it is collected after execution, please clarify whether it is euthanasia and the method of euthanasia.

Line 129 - The authors used RNA as the input material, but here is described as DNA. Please confirm if there is a problem.

Line 138,184 - The author's reference genome and Illumina sequencing data are the same accession number. Is it a mistake? And please indicate the species of the reference genome. In addition, the reference genome is only the sequencing data of a liver tissue sample, which cannot be used as a reference genome, and it is not consistent with the content of this article. Please ensure that your data is correct.

Line 195-198 - The authors found a large number of DElncRNAs between Yak and other cattle species, but the expression level of all lncRNAs and the P value of DElncRNAs were not shown in the supplementary materials. Differential expression analyses leverage reproducibility between replicates to report p-values, in the absence of which there can be reduced confidence in the findings. Please add relevant supplementary materials.

Ethical Statement - Do the authors' research unit have a cooperative relationship with the Institutional Animal Care and Use Committee of Southwest Minzu University? Why use the ethical approval of this unit?

Minor:

The English should be improved.

Line 26 - "cis" and "trans" should be Italic. Please revise all "cis" and "trans" in the manuscript uniformly.

Line 179 - " β -actin" should be Italic. In addition, the genes names and Latin names of species in all manuscripts need to be italicized. Please modify them uniformly.

Line 184-185 - No information about clean data is shown in Table S2.

Line 188 - Does novel lncRNA mean lncRNA unidentified in cattle or unidentified in all species? Please clarify.

Figure 2 - The figure shows the significance analysis, but the method section does not describe the relevant analysis.

Decision letter (RSOS-200625.R0)

Dear Dr Ji,

The editors assigned to your paper ("Transcriptome analysis identified long non-coding RNAs involved in the adaption of yak to high-altitude environments") have now received comments from reviewers.

While both reviewers think the paper is of interest, both reviewers raise a number of substantive issues which will require careful consideration and revision of the manuscript. It will be important to scrutinise and address all of the points raised by the reviewers.

We would like you to revise your paper in accordance with the referee suggestions which can be found below (not including confidential reports to the Editor). Please note this decision does not guarantee eventual acceptance.

Please submit a copy of your revised paper before 16-Aug-2020. Please note that the revision deadline will expire at 00.00am on this date. If we do not hear from you within this time then it will be assumed that the paper has been withdrawn. In exceptional circumstances, extensions may be possible if agreed with the Editorial Office in advance. We do not allow multiple rounds of revision so we urge you to make every effort to fully address all of the comments at this stage. If deemed necessary by the Editors, your manuscript will be sent back to one or more of the original reviewers for assessment. If the original reviewers are not available, we may invite new reviewers.

- Data accessibility

<http://datadryad.org/submit?journalID=RSOS&manu=RSOS-200625>

- Competing interests

- Authors' contributions

All submissions, other than those with a single author, must include an Authors' Contributions section which individually lists the specific contribution of each author. The list of Authors

should meet all of the following criteria; 1) substantial contributions to conception and design, or acquisition of data, or analysis and interpretation of data; 2) drafting the article or revising it critically for important intellectual content; and 3) final approval of the version to be published.

- Acknowledgements

- Funding statement

on behalf of Professor Steve Brown (Subject Editor)
openscience@royalsociety.org

Reviewers' Comments to Author:

Reviewer: 1

Comments to the Author(s)

Xin et al. explored their transcriptome sequencing data of yak and three cattle strains, identified lncRNAs and discussed their potential biological functions. To me, the major weakness of the present study is the lack of further validation of lncRNA's function. Under the trans model, false correlations might be included in your identification. However, I understand that yak is not a model organism and experiments on yak cell lines might be impossible currently. As a bioinformatic analysis, I think this manuscript can be published, because it provides novel lncRNA information for further research on yak. However, I encourage the authors to mention this weakness in the discussion part.

Throughout the whole manuscript, the authors used the word hypoxia instead of low oxygen condition. I think it is wrong. Hypoxia means the symptoms of organism under low oxygen condition.

Line 58: delete focused

Line 60: focusing on

Line 212: Why did you found none enriched pathways using the cis model? Is it normal in other animals? any mistake in your analysis process?

Line 215: comparisons

Line 292: it is inappropriate to using the word “uptake”

Line 294: what do you mean “consistent”?

Line 296: revised as “yak is a native species to high-altitude environments”

Reviewer: 2

Comments to the Author(s)

This article described the expression profile of lncRNAs in the gluteus tissues of Sanjiang cattle, Tibetan cattle, Holstein cattle and yak. And authors screened the potential functions and pathways of differential expression lncRNAs between yak and other cattle species. As a catalog of new transcripts, this contributes a portion of the available data. However, some problems with methods and data will affect data reliability.

Major:

Line 111 - In "2.1 Sample preparation", since the four species of cattle are raised by local farmers, apart from the differences in the transcriptome of the gluteus tissues caused by species, is it possible that the differences caused by nutritional status?

Line 113 - In "2.1 Sample preparation", please add related methods of gluteal muscle tissue collection after “All the involved animals were 60-month old”. For example, whether it is collected in vivo, and how to treat the wound after collection. If it is collected after execution, please clarify whether it is euthanasia and the method of euthanasia.

Line 129 - The authors used RNA as the input material, but here is described as DNA. Please confirm if there is a problem.

Line 138,184 - The author's reference genome and Illumina sequencing data are the same accession number. Is it a mistake? And please indicate the species of the reference genome. In addition, the reference genome is only the sequencing data of a liver tissue sample, which cannot be used as a reference genome, and it is not consistent with the content of this article. Please ensure that your data is correct.

Line 195-198 - The authors found a large number of DElncRNAs between Yak and other cattle species, but the expression level of all lncRNAs and the P value of DElncRNAs were not shown in the supplementary materials. Differential expression analyses leverage reproducibility between replicates to report p-values, in the absence of which there can be reduced confidence in the findings. Please add relevant supplementary materials.

Ethical Statement - Do the authors' research unit have a cooperative relationship with the Institutional Animal Care and Use Committee of Southwest Minzu University? Why use the ethical approval of this unit?

Minor:

The English should be improved.

Line 26 - “cis” and “trans” should be Italic. Please revise all "cis" and "trans" in the manuscript uniformly.

Line 179 - “ β -actin” should be Italic. In addition, the genes names and Latin names of species in all manuscripts need to be italicized. Please modify them uniformly.

Line 184-185 - No information about clean data is shown in Table S2.

Line 188 - Does novel lncRNA mean lncRNA unidentified in cattle or unidentified in all species? Please clarify.

Figure 2 - The figure shows the significance analysis, but the method section does not describe the relevant analysis.

Author's Response to Decision Letter for (RSOS-200625.R0)

See Appendix A.

RSOS-200625.R1 (Revision)

Review form: Reviewer 1

Is the manuscript scientifically sound in its present form?

Yes

Are the interpretations and conclusions justified by the results?

Yes

Is the language acceptable?

Yes

Do you have any ethical concerns with this paper?

No

Have you any concerns about statistical analyses in this paper?

No

Recommendation?

Accept as is

Comments to the Author(s)

The authors have resolved all my concerns. I think it is acceptable for publication.

Review form: Reviewer 2

Is the manuscript scientifically sound in its present form?

Yes

Are the interpretations and conclusions justified by the results?

Yes

Is the language acceptable?

Yes

Do you have any ethical concerns with this paper?

Yes

Have you any concerns about statistical analyses in this paper?

No

Recommendation?

Accept with minor revision (please list in comments)

Comments to the Author(s)

Line 111-113 - In "2.1 Sample preparation", the author claimed that the animals were executed by local farmers. Does the author go to the farmers' residence before execution and collect the gluteal muscles immediately after execution? In addition, since all animals are raised by local farmers, how can the author determine that all animals are 60-month old?

Line 195-198 - There is no |fold change| and Q value for differentially expressed lncRNA in the supplementary file. Please provide valid files.

Ethical Statement - If the animal is not executed by the author, there is no need for a moral statement. Because the way local farmers execute animals does not follow the Animal Care and Use Committee of Southwest Minzu University. If the author instructs local farmers to execute animals according to the guidelines, please explain.

Line 171 - "change fold" should be "fold change".

Decision letter (RSOS-200625.R1)

Dear Dr Ji

On behalf of the Editors, we are pleased to inform you that your Manuscript RSOS-200625.R1 "Transcriptome analysis identified long non-coding RNAs involved in the adaption of yak to high-altitude environments" has been accepted for publication in Royal Society Open Science subject to minor revision in accordance with the referees' reports. Please find the referees' comments along with any feedback from the Editors below my signature.

We invite you to respond to the minor comments of reviewer 2 and revise your manuscript. Below the referees' and Editors' comments (where applicable) we provide additional requirements. Final acceptance of your manuscript is dependent on these requirements being met. We provide guidance below to help you prepare your revision.

Please submit your revised manuscript and required files (see below) no later than 7 days from today's (ie 27-Aug-2020) date. Note: the ScholarOne system will 'lock' if submission of the revision is attempted 7 or more days after the deadline. If you do not think you will be able to meet this deadline please contact the editorial office immediately.

Kind regards,
Royal Society Open Science Editorial Office
Royal Society Open Science

on behalf of Professor Steve Brown (Subject Editor)
openscience@royalsociety.org

Reviewer comments to Author:

Reviewer: 1

Comments to the Author(s)

The authors have resolved all my concerns. I think it is acceptable for publication.

Reviewer: 2

Comments to the Author(s)

Line 111-113 - In "2.1 Sample preparation", the author claimed that the animals were executed by local farmers. Does the author go to the farmers' residence before execution and collect the gluteal muscles immediately after execution? In addition, since all animals are raised by local farmers, how can the author determine that all animals are 60-month old?

Line 195-198 - There is no |fold change| and Q value for differentially expressed lncRNA in the supplementary file. Please provide valid files.

Ethical Statement - If the animal is not executed by the author, there is no need for a moral statement. Because the way local farmers execute animals does not follow the Animal Care and Use Committee of Southwest Minzu University. If the author instructs local farmers to execute animals according to the guidelines, please explain.

Line 171 - "change fold" should be "fold change".

===PREPARING YOUR MANUSCRIPT===

If you have been asked to revise the written English in your submission as a condition of publication, you must do so, and you are expected to provide evidence that you have received language editing support. The journal would prefer that you use a professional language editing service and provide a certificate of editing, but a signed letter from a colleague who is a native

speaker of English is acceptable. Note the journal has arranged a number of discounts for authors using professional language editing services (<https://royalsociety.org/journals/authors/benefits/language-editing/>).

===PREPARING YOUR REVISION IN SCHOLARONE===

-- If you have uploaded ESM files, please ensure you follow the guidance at <https://royalsociety.org/journals/authors/author-guidelines/#supplementary-material> to include a suitable title and informative caption. An example of appropriate titling and captioning may be found at [https://figshare.com/articles/Table_S2_from_Is_there_a_trade-off_between_peak_performance_and_performance_breadth_across_temperatures_for_aerobic_sc](https://figshare.com/articles/Table_S2_from_Is_there_a_trade-off_between_peak_performance_and_performance_breadth_across_temperatures_for_aerobic_scope_in_teleost_fishes_/3843624) ope_in_teleost_fishes_/3843624.

Author's Response to Decision Letter for (RSOS-200625.R1)

See Appendix B.

Decision letter (RSOS-200625.R2)

Dear Dr Ji,

It is a pleasure to accept your manuscript entitled "Transcriptome analysis identified long non-coding RNAs involved in the adaptation of yak to high-altitude environments" in its current form for publication in Royal Society Open Science.

on behalf of Professor Steve Brown (Subject Editor)
openscience@royalsociety.org

Appendix A

Reviewers' Comments to Author:

Reviewer: 1

Comments to the Author(s)

Xin et al. explored their transcriptome sequencing data of yak and three cattle strains, identified lncRNAs and discussed their potential biological functions. To me, the major weakness of the present study is the lack of further validation of lncRNA's function. Under the trans model, false correlations might be included in your identification. However, I understand that yak is not a model organism and experiments on yak cell lines might be impossible currently. As a bioinformatic analysis, I think this manuscript can be published, because it provides novel lncRNA information for further research on yak. However, I encourage the authors to mention this weakness in the discussion part.

Throughout the whole manuscript, the authors used the word hypoxia instead of low oxygen condition. I think it is wrong. Hypoxia means the symptoms of organism under low oxygen condition.

Re: we have corrected the error.

Line 58: delete focused

Re: deleted.

Line 60: focusing on

Re: corrected.

Line 212: Why did you found none enriched pathways using the cis model? Is it normal in other animals? any mistake in your analysis process?

Re: We have checked the whole analysis process and we confirmed that there was no mistake. The results indicated that cis model might not be dominant in yak.

Line 215: comparisons

Re: "comparison" has been revised as "comparisons".

Line 292: it is inappropriate to using the word "uptake"

Re: "uptake" has been changed to "consumption".

Line 294: what do you mean "consistent"?

Re: "consistent" has been replaced by "constant".

Line 296: revised as "yak is a native species to high-altitude environments"

Re: revised accordingly.

Reviewer: 2

Comments to the Author(s)

This article described the expression profile of lncRNAs in the gluteus tissues of Sanjiang cattle,

Tibetan cattle, Holstein cattle and yak. And authors screened the potential functions and pathways of differential expression lncRNAs between yak and other cattle species. As a catalog of new transcripts, this contributes a portion of the available data. However, some problems with methods and data will affect data reliability.

Major:

Line 111 - In "2.1 Sample preparation", since the four species of cattle are raised by local farmers, apart from the differences in the transcriptome of the gluteus tissues caused by species, is it possible that the differences caused by nutritional status?

Re: The local farmers regularly kill animals to sell meat. At each time, dozens of animals were sacrificed. Thus, we have a choice to select individuals with similar nutritional status. We have revised the text to clarify this point.

Line 113 - In "2.1 Sample preparation", please add related methods of gluteal muscle tissue collection after "All the involved animals were 60-month old". For example, whether it is collected in vivo, and how to treat the wound after collection. If it is collected after execution, please clarify whether it is euthanasia and the method of euthanasia.

Re: The local farmers killed the animals to sell meat. After anesthetizing using electrocution, animals were killed using a high voltage. Thus, we did not treat the wound after collection. We have revised this part to clarify these details.

Line 129 - The authors used RNA as the input material, but here is described as DNA. Please confirm if there is a problem.

Re: This is correct. During the preparation of sequencing library, RNA were reverse transcribed to cDNA and then double-stranded DNA libraries were constructed.

Line 138,184 – The author's reference genome and Illumina sequencing data are the same accession number. Is it a mistake? And please indicate the species of the reference genome. In addition, the reference genome is only the sequencing data of a liver tissue sample, which cannot be used as a reference genome, and it is not consistent with the content of this article. Please ensure that your data is correct.

Re: Sorry for the incorrect accession numbers. The accession number of yak reference genome is AGSK00000000 and that of our sequence data is PRJNA512958.

Line 195-198 - The authors found a large number of DElncRNAs between Yak and other cattle species, but the expression level of all lncRNAs and the P value of DElncRNAs were not shown in the supplementary materials. Differential expression analyses leverage reproducibility between replicates

to report p-values, in the absence of which there can be reduced confidence in the findings. Please add relevant supplementary materials.

Re: All these data were supplemented in the Supplementary dataset.

Ethical Statement - Do the authors' research unit have a cooperative relationship with the Institutional Animal Care and Use Committee of Southwest Minzu University? Why use the ethical approval of this unit?

Re: The first affiliation of this manuscript does not have Ethics Committee. Three authors of this manuscript are working in the Southwest Minzu University. Thus, we submitted our protocol to the Animal Care and Use Committee of Southwest Minzu University for approval.

Minor:

The English should be improved.

Re: We have sought help from a native speaker to proof-read the manuscript.

Line 26 – “cis” and “trans” should be Italic. Please revise all "cis" and "trans" in the manuscript uniformly.

Re: the format of these words has been revised.

Line 179 – “ β -actin” should be Italic. In addition, the genes names and Latin names of species in all manuscripts need to be italicized. Please modify them uniformly.

Re: the format of these words has been revised.

Line 184-185 - No information about clean data is shown in Table S2.

Re: Number of clean reads, total bases, GC content, Q20 and Q30 are shown in Table S2.

Line 188 - Does novel lncRNA mean lncRNA unidentified in cattle or unidentified in all species? Please clarify.

Re: Novel lncRNA mean lncRNA unidentified in all species. We have mentioned this issue in the revised manuscript.

Figure 2 - The figure shows the significance analysis, but the method section does not describe the relevant analysis.

Re: The method for statistical analyses was supplemented in the revised manuscript.

Appendix B

Reviewer comments to Author:

Reviewer: 1

Comments to the Author(s)

The authors have resolved all my concerns. I think it is acceptable for publication.

Reviewer: 2

Comments to the Author(s)

Line 111-113 - In "2.1 Sample preparation", the author claimed that the animals were executed by local farmers. Does the author go to the farmers' residence before execution and collect the gluteal muscles immediately after execution? In addition, since all animals are raised by local farmers, how can the author determine that all animals are 60-month old?

Re: Yes. We went to the site and immediately collected samples after killing the animals. Most animals in the farms have clear record of birthday.

Line 195-198 - There is no |fold change| and Q value for differentially expressed lncRNA in the supplementary file. Please provide valid files.

Re: \log_2 |fold change|, p value and adjusted P values (corresponding to Q values) of DE lncRNA were described in the files of supplementary dataset.xlsx.

Ethical Statement - If the animal is not executed by the author, there is no need for a moral statement. Because the way local farmers execute animals does not follow the Animal Care and Use Committee of Southwest Minzu University. If the author instructs local farmers to execute animals according to the guidelines, please explain.

Re: Samples used in the present study were purchased from the local farmers. The animals were sacrificed by the farmers following the method described in the Materials and Methods section. No special ethical statement is required for the present study.

Line 171 – “change fold” should be “fold change”.

Re: revised accordingly.